# Emerging Diagnostics in *Clostridioides difficile* Infection

**DOI:** 10.3390/ijms25168672

**Published:** 2024-08-08

**Authors:** John P. Hulme

**Affiliations:** Department of Bio-Nano Technology, Gachon University, Seongnam-si 13120, Republic of Korea; flp15@gachon.ac.kr

**Keywords:** endospores, automated, toxins, point of care, CRISPR, mass spectrometry

## Abstract

*Clostridioides difficile* detection in community settings is time-intensive, resulting in delays in diagnosing and quarantining infected individuals. However, with the advent of semi-automated devices and improved algorithms in recent decades, the ability to discern CDI infection from asymptomatic carriage has significantly improved. This, in turn, has led to efficiently regulated monitoring systems, further reducing endemic risk, with recent concerns regarding a possible surge in hospital-acquired *Clostridioides difficile* infections post-COVID failing to materialize. This review highlights established and emerging technologies used to detect community-acquired *Clostridioides difficile* in research and clinical settings.

## 1. Introduction

First isolated in 1935 from the stools of newborn infants, the environmentally ubiquitous anaerobe *Clostridioides difficile* (*C. difficile*) is a spore-forming Gram-positive bacillus with toxigenic potential [1]. The rod-shaped pathogen secretes two toxins, enterotoxin A (tcdA) and cytotoxin B (tcdB), which are primarily responsible for the pathogenicity observed in individuals with *C. difficile* infection. In addition to tcdA and tcdB, there is a third binary toxin (CDTa and CDTb), denominated CDT, which is thought to upregulate tcdA and tcdB production [1,2]. The role of CDT is unclear, although initial experiments suggest it may enhance pathogenic virulence by suppressing the protective colonic eosinophilia in the host [3]. Other virulence factors include surface-layer (Cwp84 and 66) and flagella (FliC and FliD) proteins associated with bacterial adherence and migration [4]. The inflammatory role of these toxins and associated regulators (tcdC) is well documented and is beyond the scope of this review [1].

*C. difficile* infections (CDIs) can be characterized as either community-acquired (CA-CDI) or healthcare-acquired (HA-CDI), depending on whether symptoms present in the first 3 days upon hospitalization or later [5]. Delineating HA-CDI from CA-CDI transmission remains challenging; initial studies involving 957 patients identified 333 isolates with 2 single-nucleotide variants (SNVs) and 488 with 10 SNVs. Of the 333 individuals, 35% had a clonal relationship with the same ribotype, whilst 120 exhibited no identifiable epidemiological link [6]. Furthermore, isolates with the same genotype showed substantial genetic diversity, suggesting that many unknown reservoirs (asymptomatic carriers and hospital staff) may have contributed to hospital transmission. Asymptomatic carriage can be defined as the presence or detection of *C. difficile* toxins in the absence of known disease pathologies. In addition [7] to asymptomatic carriers, other studies showed that a large percentage (27%) of HA infections and returning (previously *C. difficile*-negative) patients were community-acquired and subsequently imported into the hospital [8].

The risk factors for HA-CDI and CA-CDI in adults include prior antibiotic use [9,10], chronic inflammatory diseases [11], gastric acid medications, nutrient availability, and reduced microbial diversity [12]. Notably, 31% of younger, healthy UK females (median age: 50–51) without prior antibiotic exposure were diagnosed with CA-CDI [13]. Recent reports suggest that family exposure is a significant risk factor for increased incidence [14]. Younger family members are particularly susceptible, with prevalence in children (<16 years) approaching 40% and 53% in neonates and infants (2–12 months), with many being asymptomatic [15]. Recurrence in children mirrors adults at 20–30%. Factors contributing to neonatal susceptibility include nutritional deficiencies, reduced colonization resistance, and inadequate anamnestic immune responses [16,17].

In addition to human hosts, whole-genome sequencing (WGS) data indicate that farm animals, such as pigs and chickens, can act as reservoirs for *C. difficile*, facilitating zoonotic bidirectional transfer (ZBT) [18]. *C. difficile* is also found in rivers, lakes, soil, household environments, and the nasal–oral airways of pets like dogs and cats [19,20,21,22]. Phylogenetic analysis shows extensive clustering of human, animal pathogenic, and antibiotic-resistant *C. difficile* strains, highlighting the role of farm and food networks in long-distance transmission [23]. Sequence type 11 (ST11) ribotypes, including hypervirulent RT027, have multiple tetM-associated clonal expansions [24]. Of note are the resistant alleles, tetM 10 and tetM 16, shared among pathogenic and zoonotic species [25,26]. As of 2020, over 800 *C. difficile* ribotypes have been reported, with several frequently identified in the USA [27]. *C. difficile* endospores are highly resilient, surviving various stressors inside and outside hosts until conditions allow for germination [28,29,30,31,32,33]. Various transmission pathways for CD endospores in the mammalian infection cycle are shown in Figure 1 [34].

Given the ubiquitous nature of *C. difficile* in food and medical settings, recent concerns have focused on identifying hypervirulent strains (RT027) using biochemical, molecular, and immunological techniques [1,35,36,37]. Rapid testing methods like PCR tests provide results within hours. Enzyme immunoassays and lateral flow assays are now more user-friendly and cost-effective, suitable for multiple settings [38,39,40,41].

## 2. Identification of Resistant and Viable Spores

The numerous challenges in monitoring the emergence of new and recurring infections have led to a surge in research identifying the aerotolerant properties of CD spores and the conditions governing their transmission, germination, and inhibition. A key concern regarding transmission is spore shedding, which occurs in clinically infected patients, asymptomatic carriers, and up to 56% of patients who have completed a CDI-directed antibiotic course after four weeks [42,43,44]. The germination of spores left over in the GI tract after antibiotic cessation and the acquisition of different strains from the environment are thought to be the main causes of recurring infections (rCDIs).

Under a scanning electron microscope, metabolically dormant spores appear as multilayered composites with an outer exosporium affording a high degree of resilience to primary stressors (pressure and temperatures), while the inner layers protect the DNA and associated proteins from secondary insults until germination conditions present. Germination is initiated by the primary bile acid taurocholate, which is detected by the pseudoprotease CspC, eventually leading to the activation of the cortex lytic hydrolase SleC and spore rehydration. A schematic of the spore structure and main and co-germinates is shown in Figure 2. For those readers interested in a more detailed explanation of the molecular processes regulating sporulation and germination, an excellent review is recommended [45].

The high infectivity and considerable management expenses associated with CDI are mainly due to the ability to produce resilient endospores. These spores are abundant in animal fecal matter (stage II) and readily adhere to various inanimate surfaces in aerobic and anaerobic environments (hospitals, slurry farms, and water treatment facilities). To effectively combat these spores, it is critical to understand their resilience mechanisms and how they respond to chemical agents (sodium hypochlorite, glutaraldehyde, or peracetic acid). In cases of severe persistence, methods like ultraviolet radiation (UV-C), steam, hydrogen peroxide aerosol, and ozone have been employed for their sporicidal effectiveness.

Sporicidal effectiveness on inanimate surfaces can be assessed using various non-invasive methods such as Raman and infrared spectroscopy (RS and IS). Compared to infrared [46], RS produces sharper signal bands and is less sensitive to signal interferents (oxygen levels), rendering it applicable to dry and wet measurements. These advantages have been employed in identifying different spore strains [32] and assessing the structural resilience of hypervirulent spores (R20291) to biocides. For example, recent work by Malyshev et al. used a modified version of RS, namely laser tweezer Raman spectroscopy (LTRS), to examine the resilience of three strains (DS1813, CD630, R20291) to sodium hypochlorite. LTRS generates a spectral fingerprint of the trapped object, enabling the detection of chemical differences between treated and untreated spores and distinctions within vegetative cells. A key indicator of internal spore damage is a loss of structural integrity, which presents a reduction in the intensity of the CaDPA Raman peaks. Upon exposure to 0.5% hypochlorite (aq) (hospital surface decontamination requirements, 4 log reduction), CD630 and R20291 exhibited 2.2 ± 0.1 and 0.8 ± 0.1 log reductions compared to controls. The researchers attributed the loss in sensitivity to mechanistic differences and reduced surface area (spore aggregation). Such a loss in R20291 sensitivity was unexpected, with authors citing the overuse of chlorine-releasing agents (CRAs) during the pandemic as a potential cause worthy of future investigation.

DPA leakage and its chelating potential can also be assessed using turbidity, absorbance, scattering, and fluorescence spectroscopic measurements [47,48,49]. For example, surface-enhanced Raman scattering (SERS) was recently combined with functionalized silver–terbium nanoparticles specific to DPA in the simple and sensitive detection of spores [50]. Another approach employed the enhanced luminescence from a terbium–lanthanide–fluorescent polyfluorene (PFO) semiconductor complex upon exposure to CaDPA, exhibiting a detectable concentration range of 2.5 to 25 nM for the chelating agent [51]. An alternative trivalent ion to terbium is europium, which was recently incorporated with a metal–organic gel (MOG), producing an enhanced electrochemiluminescence (ECL) signal upon exposure to DPA with a detection limit of 7.35 nm [52]. Terbium and europium are expensive; thus, divalent substitutes like magnesium and copper are also employed. Recent work using a catechol-substituted monostyryl boradiazaindacene (BODIPY) by Cetinkaya et al. [53] to sequentially detect Cu^2+^ ions and dipicolinic acid (DPA) permitted the detection of low concentrations of spores 1.0 × 10^5^/mL in solution. For those researchers interested in investigating the strain-specific properties of bacterial spores via LTRS and other optical spectroscopic techniques, the review by Malyshev et al. is suggested [54].

Apart from DPA leakage, the structural viability and adherence of resistant spores on organic and inorganic surfaces can be assessed via various forms of microscopy [55]. Understanding the properties that enhance spore adherence is crucial for reducing transmission and preventing infection recurrence. In a recent study [32], the viability of *C. difficile* spores (DS1748, R20291, and DS1813), seeded on a range of clinically relevant surfaces such as isolation gowns, stainless steel, and floor vinyl, was assessed after exposure to a high-strength biocide (sodium dichloroisocyanurate). All strains remained viable on the clinical surfaces after exposure to the recommended disinfection concentration, demonstrating ineffectual sporicidal action. An interesting observation from this study was the easy exchange of spores between hydrophobic surfaces, such as fluid-resistant gowns and stainless steel. Of notable concern was the inability of single-use gowns to trap spores within their fibers and prevent onward transmission.

In addition to surface adherence, *C. difficile* spores can survive in abiotic environments by embedding themselves in mono- or mixed-species biofilms [56]. These biofilms serve multiple roles that may contribute to recurrence. They can act as reservoirs for spores and vegetative cells, shielding them from antibiotic activity and promoting persistent cell formation. In a recent study [57], researchers compared the differences between vegetative cells and spore counts as well as changes in the biomass of biofilms after exposure to disinfectants using optical density measurements, inverted light, and confocal laser scanning microscopy. In evaluating the effectiveness of hospital-based disinfectants on *C. difficile* embedded in biofilms, it was found that no disinfectant could eliminate *C. difficile*. However, generic disinfectants, Clorox, OPA, and Virex, were the most effective at killing *C. difficile* spores, irrespective of the age, ribotype, or wash condition.

The germination of *C. difficile* spores into vegetative form within the mammalian host (Figure 1, stage III) encompasses the major cellular structure, machinery, and morphology alterations, resulting in significant changes to the electrophysiological state of the spore. Assessment of *C. difficile* spore germination usually involves measuring colony-forming units (CFUs), which is labor-intensive and takes at least 24 h but is necessary due to high recurrence rates of nosocomial antibiotic-associated diarrhea [58]. The structural transition (changes in interior conductivity and membrane resistance) from spores to the vegetative form can be monitored via dielectrophoresis (DEP). Dielectrophoresis involves the movement of polarized cells within a spatially non-uniform electric field and has been used to measure differences in bacteria based on cell walls [59]. Recently, a rapid method using high-throughput single-cell impedance cytometry (>300 events/s) was developed to quantify live bacterial cells by distinguishing their electrophysiology from spores in vivo (mouse model) and in vitro. This method allowed germination assessment after just 4 h of culture, with a detection limit of ~100 live cells per 50 μL sample [60].

*C. difficile* strains with increased sporulation are associated with increased transmission rates [42]. Methods quantifying sporulation characteristics vary greatly, relying on small sample sizes preventing standardizations [58]. Burns et al. estimated that the sample size should employ at least seven strains of a given type (BI/NAP1/027), suggesting there is a lack of evidence to support the claim that epidemic *C. difficile* types form spores with greater efficiency. In an effort to improve the consistency of further studies, the authors presented an experimental design capable of independently assessing four factors (vegetative cell growth rate, sporulation rate, total sporulation, and persistent vegetative cells) via a combination of methodologies. Such an approach was employed in the assessment of octahedron iron oxide nanocrystals to inhibit the germination and viability of two strains of *C. difficile* (CCUG 37780 (tcdA−, tcdB−) and CCUG 19126 (tcdA+, tcdB+)) in a mouse model [61]. The nanoparticle inhibited germination in a size/dose-dependent manner, although there was some strain variation. Coupled with the superior biocidal (comparable to bleach) activity, the authors suggested that the nanocrystals block the germinant receptors, altering the surface properties (change hydrophobicity) and thus preventing germination, which is a similar theory posited by similar authors in another report [62].

When the surface of a spore is defective (absence of exosporium, slow CaDPA release or mutation), established techniques may be insufficient; in such circumstances, an automated time-lapse microscopy-based germination assay may be better suited. In a recent study, a time-lapse microscopy (TLM) pipeline was used to observe morphological changes and stages of germination in real time using mutant spores [63]. Researchers concluded that Ca^2+^ is not essential for initiating spore germination and that CaDPA can enhance the germination of nearby spores via a feedforward loop. In addition to TLM, a protocol for quantifying the germination properties of individual bacterial endospores using a Python program (PySpore) was utilized to compensate for germination heterogeneity within a population [63]. PySpore allowed researchers to deconvolve heterogeneity in the time to germination and germination rate, which could not be distinguished in bulk analyses. Another approach to spore heterogeneity and variations in methodologies is the utilization of digital microdroplet platforms in single-spore [64] sampling and analysis, as recently proposed by Bernier et al. [65]. The advantages of microdroplet technology in biological research (engineered strain modification and mutant host screening) are well documented [66,67]. For example, Tu et al. [68] presented an optimized platform with a sorting capacity close to 10,000 *Streptomyces lividans* 66 variants per hour with an enrichment ratio of up to 334.2. When combined with fluorescent activated sorting (FADS), the results were shown to be consistent with those from 24-h cultivation microtiter plates.

## 3. Sampling the Community (Ribotyping and Mass Spectrometry)

PCR-ribotyping has been used as the reference standard to investigate transmission in suspected CDI outbreaks for decades. The method analyzes the varying intergenic spacer regions (ISRs) between the 16S and 23S rRNA genes. PCR across the ISR generates fragments of different lengths, the sizes of which can be resolved by gel or capillary electrophoresis [69]. The variability of the spacer regions between strains has been used to define hypervirulent lineages and trace epidemiological links, notably RT027 and RT078. However, PCR-ribotyping shows a low discriminatory capability in differentiating strains having ISRs of equal length but different nucleotide sequences [70]. Other *C. difficile* typing methods include pulsed-field gel electrophoresis (PFGE), multi-locus sequence typing (MLST), whole-genome sequencing (WGS), multi-locus variable-number tandem repeat analysis (MLVA), and matrix-assisted laser desorption ionization time of flight (MALDI-TOF) mass spectrometry.

Mass spectrometry (MS) techniques such as gas chromatography–MS (GC-MS), high-resolution tandem MS (LC-MS/MS), and liquid chromatography–MS (LC-MS) are increasingly employed for ribotyping pathogenic bacteria in clinical samples. Soft ionization methods, including matrix-assisted laser desorption ionization (MALDI-TOF-MS) and electrospray ionization (ESI)-MS, allow the analysis of large molecules, whole cells, proteins, DNA, and, recently, the parasite *Plasmodium falciparum* in human blood [71].

MALDI-TOF is particularly effective for directly analyzing bacterial isolates, delivering identification results in approximately 15–30 s. In clinical microbiology, MALDI-TOF MS produces a protein spectrum of a sample that is sourced to a database/reference library, where potentially new specific biomarkers are identified. This combinatorial approach facilitates microbial protein identification through peptide mass fingerprinting and the analysis of nucleic acid sequences and amplification products. In 2021, researchers evaluated the possibility of differentiating and classifying the different toxigenic *C. difficile* strains found in 158 patient fecal samples using MALDI-TOF MS and two statistical classifying algorithm models (CAMs) [70]; the first CAM involved all 10 ribotypes, whereas the second one only involved 5 (PR1–PR5) ribotypes. Improved performance was achieved with the second CAM, showing a 100% recognition capability, 96.6% cross-validation accuracy, and 98.4% agreement (60 correctly typed strains, classified as PR1–PR5, out of 61 examined strains) with PCR-ribotyping results. The following year, the same group used MALDI-TOF MS to characterize an outbreak of 40 toxigenic *C. difficile* DNA-positive patients in an Italian hospital [72]. In agreement with PCR, four strains were classified as RT 126 by T-MALDI. Of the 29 strains identified, 22 were linked to the outbreak, belonging to the same cluster (PRA). Further analysis revealed that 22/29 were tcdA+/tcdB−/cdt+, 4/29 were tcdA+/tcdB+/cdt+, and 3/29 strains were tcdA+/tcdB+/cdt−. The authors noted that the A+ B-phenotype is mostly encountered in food animals [15].

MALDI-TOF-MS is a typing method that is readily amenable to the epidemiological tracking of emerging and hypervirulent strains of *C. difficile* and has been employed in the identification of numerous RTs such as RT001, RT017, RT027/RT176, and RT078/RT126 [73,74]. As well as CAMs, MALDI-TOF-MS was recently paired with four machine learning models and 65 isolates as a training set for species identification and the discrimination of hypervirulent (HV) from non-hypervirulent (NHV) ribotypes [75]. The authors showed that protein extract-based MALDI-TOF spectra coupled with ML could distinguish between HV and NHV ribotypes circulating in Europe (accuracy > 95%). Furthermore, subtyping of certain HVRTs (e.g., RT027/176 or RT023) was possible (100% accuracy, PLS-DA model).

Whether using ML or a CAM, developing and training these tools necessitates exposure to many different ribotypes to accurately and specifically classify strains of clinical and epidemiological significance. Thus, it may be argued that machine learning paired with MALDI-TOF MS could be a powerful tool for epidemiological surveillance centers involved in tracking CDI worldwide, akin to WGS. Like WGS, better performance will require continuous updating based on epidemiological data to include new unknown circulating ribotypes.

### NAAT and CRISPR-Cas12a (Toxin Identification)

NAAT provides the most useful diagnostic information [76], offering several advantages, including excellent sensitivity, low complexity, simplified reporting, and a reduced need for repeat testing. To date, the Food and Drug Administration (FDA) have approved 23 nucleic tests (https://www.fda.gov/medical-devices/in-vitro-diagnostics/nucleic-acid-based-tests#microbial, accessed on 10 June 2024). Five utilize either isothermal helicase-dependent (HDA) or loop-mediated isothermal amplification (LAMP), while the remaining assays predominantly use real-time or multiplex PCR (mPCR). The majority of NAATs target the genes for tcdB, tcdA, and/or the binary toxin in diarrheal stool specimens and have exhibited good sensitivity (80–100%) and specificity (87–99%) [77] values. Of note is the recent work by Wu et al. in which a new single-tube multiplex real-time PCR method for simultaneously detecting tcdA, tcdB, and cdtB targets in 74 fecal samples was reported [78]. The method demonstrated an intraspecies specificity of 100%, a consistency value of 98.4%, and similar sensitivities compared with RT-PCR. Moreover, the technique identified toxigenic *C. difficile* in 54 fecal samples, exhibiting sensitivities and specificities of 96.49% and 94.12%, respectively.

PCR can only detect toxin-encoding genes rather than directly detecting the toxins in stool, which may result in the overdiagnosis of CDI, potentially ruling out other causes of diarrhea [79]. Several PCR panels/platforms have sought to address this concern, such as Luminex xTAG GPP (Luminex Molecular Diagnostics Inc., Toronto, ON, Canada) and FilmArray™ Gastrointestinal Panel (BioFire Diagnostics, Salt Lake City, UT, USA), which can detect multiple microbial pathogens (seven bacteria, two viruses, and two parasites, and fourteen bacteria, five viruses, and four parasites), including *C. difficile*, frequently encountered in stool samples. Regarding the former platform, the sensitivity ranges from 90% to 100% (depending on the target organism), with an overall sensitivity of 94.5% and a specificity of 99.1% being reported [80]. A multicenter trial using the BioFire panel identified 1180 potential pathogens in 1556 specimens tested, demonstrating 100% sensitivity for 12/22 targets and 94.5% for an additional 7/22 targets [81]. More recently, a direct comparison between the two showed that samples with low-concentration targets were more often detected by FilmArray than with Luminex xTAG GPP, and xTAG GPP was more likely to be affected by amplification inhibitors [82].

While employing multiple targets offers benefits, MPCR is still vulnerable to proteinous interferents, which can delay or elevate the C_T_ (threshold cycle) and lead to inaccurately low template estimates [83]. The benefits of an established or fixed C_T_ value were demonstrated in a large study by Garvey et al. using the GeneXpert assay with C_T_ values correlating with toxin positivity. In addition, the researchers found that a lower CT correlated with a higher failure rate with first-line therapy and higher mortality in patients [84]. Conversely, several smaller studies found significant overlap in C_T_ values between symptomatic patients and asymptomatic carriers, making it unreliable to use C_T_ values to distinguish disease from colonization [85].

Another type of multiplex PCR akin to LAMP that does not use thermal cycles and functions at room temperature (target DNA at 37–42 °C) is recombinase polymerase amplification (RPA). The properties of RPA partly meet the need for a rapid, inexpensive, ultra-sensitive POCT for detecting *C. difficile* in hospital admission and community settings, where distinguishing symptomatic patients/hosts from asymptomatic carriers is critical. In regard to the ultra-sensitivity requirements for a POCT, the attomole precision of a clustered, regularly interspaced short palindromic repeat (CRISPR)/Cas 12a diagnostic may provide a solution [86]. Cas12a, a CRISPR type V system component, induces the cleavage of non-targeted single-stranded DNA (ssDNA). This activity facilitates the detection of nucleic acids via signal amplification and supports various readouts by incorporating functionalized reporter nucleic acids. Cas12 enzymes require a protospacer-adjacent motif (PAM) site in the target region for dsDNA cleavage to collaterally cleave ssDNA [87]. Cas12a-based detection has been successfully applied to various pathogens, including SARS-CoV-2 and *Salmonella* [88,89]. Recent work by Shen et al. combined the advantages of both RPA and Cas12a [90] in detecting tcdA and tcdB via two multiplex RPA-Cas12a-assay formats, namely camera capture UV fluorescence and a visible lateral flow strip (LFS). The strain specificities of both formats were validated against the extracted DNA of 14 diarrhea-causing enteric pathogens, and clinical potential was verified via a commercially available A/B PCR kit. Regarding sensitivity, the authors reported that both formats were equally consistent, exhibiting limit of detection values of 10 copies/μL and 1 copy/μL for tcdA and tcdB, respectively.

## 4. Established and New Immunological and Enzyme Assays

At the start of germination, multiple metal storage mechanisms at the host epithelial interface act to starve bacteria of nutrient metals in a process known as nutritional immunity [91]. As a countermeasure, the anaerobe releases a battery of toxins and digestive enzymes, the degree to which (depending on the ribotype) present in the host’s or patient’s stool. Detection of these proteins can be achieved through various methods, such as nucleic acid amplification tests (NAATs), glutamate dehydrogenase (GDH) enzyme immunoassays (EIAs) for toxins A and B and/or toxinogenic culture (TC), and a cell cytotoxicity neutralization assay (CCNA). Regarded as the gold standards for diagnosing CDI (*Clostridioides difficile* infection), in the first stage of TC, toxinogenic isolates are identified via media such as modified CCFA (cyclo-serine–cefoxitin–fructose agar with biliary salts and egg yolks) and ChromID *C. difficile* agar (bioMérieux, Grenoble, France). Studies show that ChromID yields a higher 24 h recovery (sensitivity, 92%) than CCFA (sensitivity, 22%) [92,93,94]. Another study confirmed that ChromID *C. difficile* agar outperforms CCFA, cyclo-serine–cefoxitin–egg yolk agar, and tryptone soy agar with sheep blood [95]. In step two, isolates are re-cultured in broth, and the supernatant is filtered and added to a cell line culture. The cytopathic effect (CPE) is evaluated and neutralized by antitoxins.

For CCNA, CPE is observed as cell rounding; some strains may induce protrusions, known as a sordellii-like CPE. If the CPE is reversed by antitoxins, the test is positive for *C. difficile* toxins. Various cell lines have been used for toxin detection, including African green monkey kidney, McCoy, MRC-5, primary rhesus monkey kidney, and Vero cells [96].

Although TC and CCNA have demonstrated high sensitivities (90–95%), they are cumbersome, costly, and time-consuming, typically requiring 3–5 days to complete. In addition, the former test is subject to pre-analytic factors and user experience; thus, occasional clinical cases of CDI may be missed [97]. Faster response times can be achieved with cheaper EIA tests, which offer 96–98% specificities but have relatively low sensitivities of 52–75% for toxins A and/or B. Similar response times are possible with GDH tests, which exhibit widely varying sensitivities (50–99%) and specificities (70–100%) [98]. In a zoonotic study, GDH sensitivity (89.4%) values have been achieved with immunochromatographic tests for *C. difficile* in dog stool samples [99]. Currently, several commercial toxin enzyme immunoassays (toxin EIAs) such as ProSpecT Toxin A/B (Remel Products, Thermo Fisher Scientific, Lenexa, KS, USA) and *C. difficile* Tox A/B II (TechLab Inc., Radford, VA, USA) as well as glutamate dehydrogenase (GDH) EIAs like *C. Diff* Chek-60 and *C. Diff* Quik Chek (TechLab Inc., Radford, VA, USA) are commercially available. These products generally have a low cost per test. While GDH EIAs offer high sensitivity of up to 90%, they cannot differentiate between CDI and asymptomatic colonization or the presence of non-toxigenic strains. An attempt to remedy the GDH diagnostic quandary was recently proposed by Liu et al. [100], in which optimized diagnostic algorithms incorporating the cost-effectiveness of GDH with high specificity and increased sensitivity were evaluated. Using 39 TC-positive and 147 TC-negative samples, the researchers assessed the performance of two algorithms, GDH-AB and GDH-NAAT. The former algorithm showed good concordance with TC in detecting toxigenic *C. difficile* (kappa = 0.82), while the sensitivity (48.7%) of the GDH-AB algorithm was too low to meet the demand for CDI diagnosis.

Several improvements in assay design have sought to address the low sensitivity of GDH-AB and the inability of standalone tests to discern asymptomatic carriage from an active infection. Of note is the recent work by Han et al. [101], in which a paper-based multiplex analytical device (mPAD) based on a lateral flow assay (LFA) was constructed with a unique configuration and architecture designed to overcome limitations in multiplexing, sensitivity, simplicity. Using signally enhanced paper-adhered gold nanoparticle-labeled antibodies, the researchers presented a multiplex device that permitted visual confirmation of GDH, toxin A, and toxin B in 32 positive and 8 negative fecal samples. For additional signal enhancement, gold ions were catalytically precipitated onto AuNPs accumulated on the test line, leading to particle size growth that allowed for better visualization of the test line and significant improvements in device performance. The device exhibited a sensitivity of 97%, specificity of 88%, accuracy of 95%, and LOD for GDH, tcd A, and tcd B of 0.16 ng mL^−1^, 0.09 ng mL^−1^, and 0.03 ng mL^−1^, respectively. The inclusion of the catalytic step and its visual effect can be seen in Figure 3.

NanoBiT immunoassays have reliably quantified several protein biomarkers and antibodies over concentrations spanning four orders of magnitude with low pM sensitivity and up to a 1000-fold signal-to-background ratio [102]. Recently, Adamson et al. utilized a POCT NanoBiT Split-Luciferase Assay to enhance GDH and Toxin B (TcdB) sensitivities [103]. NanoBiT GDH and TxB assays conducted in 3.33% (*w*/*v*) feces showed significant reductions in assay sensitivities (LOD = 4.5 and 2 pM, respectively) with overall performance values akin to the currently used point-of-care LFT, Quik Chek Complete.

In addition to tcdB and A, another protein secreted by clostridial bacteria is PPEP-1, a unique zinc metalloprotease. PPEP-1 belongs to a family of proteases that specifically cleave Pro–Pro bonds within proline-rich motifs. Its consensus sequence, distinct from other PPEP proteases, is conserved among *C. difficile* strains and is essential for pathogen mobility and gut colonization [104,105]. The unique cleavage activity of PPEP-1, exclusive to *C. difficile*, could serve as a specific and novel marker for CDI. Recent work by Prescher et al. coupled PPEP-1 activity with a “turn-on” Nluc bioluminescent sensor that could rapidly indicate CDI without requiring expensive equipment [106]. NanoLuc (Nluc), known for its exceptional brightness and stability, is commonly utilized in sensing platforms for various target biomolecules. These properties allowed for detecting PPEP-1 concentrations as low as 10 nM in fecal solution using a phone camera within minutes. For rapid diagnosis, the European Society of Clinical Microbiology and Infectious Diseases (ESCMID) recommends using a two-step algorithm that begins with either NAATs or GDH-EIA tests, as shown in Figure 4.

## 5. Ultra-Sensitive Immunoassays and Real-Time Analysis

In an attempt to discern CDI from asymptomatic carriage, many laboratories often utilize the sensitivity of an NAAT in combination with the specificity of an EIA in a multistep algorithm [107]. However, multistep algorithms are expensive and are often out of reach in rural India and Brazil [87], where a single inexpensive test is often employed. However, the current requirements of high specificity and ultra-sensitivity suggest that only a few single-molecule counting platforms, such as the single-molecule array (SIMOA) and Singulex Clarity measurement systems [108], can achieve the required level of performance. In a recent study, the Singulex Clarity *C. diff* toxin A/B assay measured TcdA and TcdB in stool samples using the Singulex Clarity system. Upon evaluating 311 patient stool samples, the Singulex Clarity *C. diff* toxin A/B assay yielded 97.7% sensitivity and 100% specificity in less than 40 min. Moreover, the automated assay exhibited a limit of detection (LOD) of 2.0 pg/mL (TcdA) and 0.7 pg/mL (TcdB), several orders of magnitude higher (800 to 2500 pg/mL) than current EIAs.

Another molecular platform is the “single-molecule array” (SIMOA), which isolates and detects individual proteins on paramagnetic beads in femtoliter-sized wells, enabling more sensitive immunoassays than conventional EIAs. Research employing SIMOA demonstrated a limit of detection (LOD) of 0.6 and 2.9 pg/mL in pretreated fecal aliquots for toxins A and B, with an intra-assay coefficient of variation below 10% [109]. The platform also detected toxins in 24% more samples with laboratory-defined CDI than the high-performing toxin EIA (95% (63/66) versus 71% (47/66)). However, further research by the previous group indicated that toxin concentration alone was not definitive for symptomatic CDI using SIMOA, suggesting that a real-time approach may be necessary [110].

As highlighted by Song et al., a simmering concern regarding the applicability of these standalone ultra-sensitive tests is the preanalytical loss of toxin activity and the inability to distinguish colonized patients from infected ones [111]. A recent study by Sandlund et al. set out to address this clinical dichotomy by comparing the performance of the Singulex Clarity (SC) *C. diff* toxins A/B with a multistep algorithm using a GDH-and-toxin EIA and toxins A and B arbitrated by a semiquantitative CCNA [112]. Upon evaluation, the researchers reported that Clarity had 96.2% negative agreement with GDH−/toxin− samples, 100% positive agreement with GDH+/toxin+ samples, and 95.3% agreement with GDH+/toxin−/CCNA− samples. In conclusion, the authors suggested that the Clarity toxin assay may offer an accurate, standalone solution for CDI diagnostics.

The real-time cell analysis (RTCA) system stands out as a label-free, real-time, and non-invasive technique with high sensitivity and specificity, and it has been employed in previous studies to quantify functional and toxical tcdB concentrations in fecal samples [113]. A recent study [114] using RTCA highlighted the significant differences in functional and toxical tcdB levels in the supernatant of *C. difficile* colonized (CDC) and CDI group samples, even though the total levels of TcdB protein were similar, 490.00 ± 133.29 ng/mL and 439.82 ± 114.66 ng/mL. Interestingly, co-culturing tcdB with CDC samples resulted in a significant reduction in toxin concentration, suggesting the RTCA process may require an extra degree of refinement (increase in ultracentrifugation speeds, additional filtration) so that potential interferents such as bile acids, outer membrane vesicles (OMVs), cellular exosomes, and C-terminal combined repetitive oligopeptides (CROPs) are excluded from assay measurements [115]. In order to identify the suspected interferents, the authors recommended conducting a study on a comparative proteome and metabolomics.

Additional efforts to overcome the limitations (inherent matrix-associated cytotoxicity (MAC) and subjective interpretation of cytopathic effects of classical CCNA), resulting in improvements in the diagnostic accuracy of CDI, were recently addressed by Elfassy et al. [116] in which an automated version of CCNA with a luminescent readout was used to analyze clinically relevant stool samples for infection. In the first instance, a viability threshold was established by determining a viability/fold that offered 98% specificity with 99.98% confidence to discern the effects of toxin B from those generated by MAC on Vero cells. With the automated assay, positive cut-offs using diarrheal samples from donors classified as CDI-negative demonstrated a minimum of 98% assay specificity. In conclusion, the authors suggested that automated CCNA should be combined with a highly sensitive molecular test and employed as either a prescreen or confirmatory test, providing a distinct advantage in detecting early infections or mild cases where the pathogen load is below the detection threshold of standard tests [117]. A selection of traditional and emerging detection methods is shown in Table 1.

Clearly, the cost of USI may limit its broader application to research settings and clinical trials involving immunocompromised and immunosenescent cohorts where low levels of toxin can be significant, particularly in periods of endemicity, where high-throughput screening is required [118,119,120]. Moreover, automated USI could also enhance elderly monitoring (care settings) by accounting for atypical symptoms and comorbidities [121], thereby further reducing the risk of CDI transmission and overdiagnosis.

## 6. Conclusions

Given their ability to persist across a network of aerobic and anaerobic environments found in community, industrial, and hospital settings, the monitoring and diagnosis of *C. difficile* endospores and vegetative cells require an integrated solution that substantially reduces community reservoirs and associated chains of transmission [9]. Increased transmission has been associated with the sporulation rates in animal and human hosts and the economic conditions (suspension of antibiotic stewardship) that favor an increase. This review initially highlighted the different spectroscopic and cytometric techniques used to examine the properties of endospores that enable their resistance to numerous environmental stressors and the microbial (dysbiosis) and chemical factors that influence germination. The surveillance potential of the combined roles of mass spectroscopy and PCR-ribotyping with machine learning in discerning epidemiological and genetic factors was also discussed.

A consistent theme throughout this review is the inability of employed molecular techniques to sufficiently differentiate between CDI and asymptomatic colonization or the presence of non-toxigenic strains. Currently, no FDA-approved single rapid test can reliably diagnose *C. difficile* disease even in the presence of good-quality fecal samples [122,123,124]. While NAATs offer excellent sensitivity for detecting toxigenic *C. difficile*, they can also identify patients colonized with *C. difficile* without toxin production, potentially leading to overdiagnosis and overtreatment. On the other hand, toxin EIAs (enzyme immunoassays) provide good clinical specificity in symptomatic patients but lack sufficient sensitivity to rule out CDI. Additionally, the presence of toxins does not correlate with true CDI in asymptomatic carriers or cured patients. Even with the advent of ultra-sensitive single molecular assays and real-time monitoring, a definitive diagnosis for CDI remains out of reach, partly due to the absence of institutional agreements and poor patient selections.

Regardless of the testing method, enhancing test utilization is essential for accurately identifying true *C. difficile* cases in animals and humans. Furthermore, to implement meaningful interventions that positively affect human and animal health, it is crucial to go beyond the hospital setting and cultivate collaborative relationships among industry, government, veterinarians, clinicians, and researchers.

## Figures and Tables

**Figure 1 ijms-25-08672-f001:**
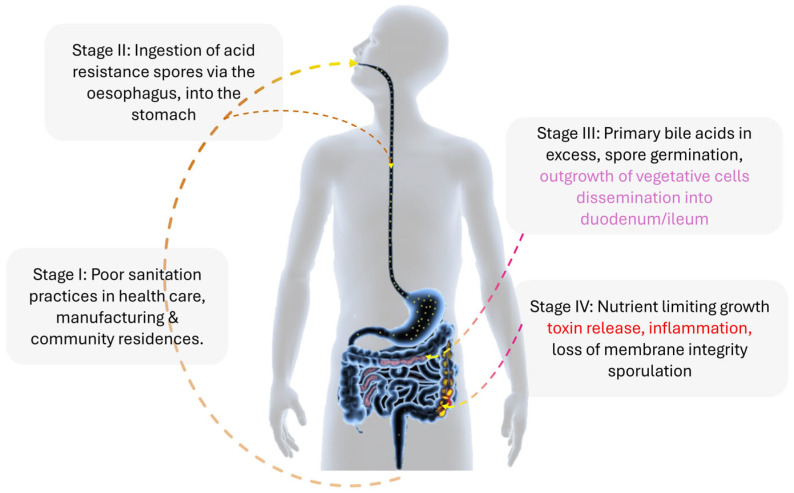
The infection cycle of *Clostridioides difficile* (*C. difficile*) in a mammalian host. *Clostridioides difficile* infection in a mammalian host begins with ingesting *C. difficile* spores (yellow spots). These spores travel through the esophagus to the stomach. From the stomach, the spores move to the duodenum and ileum, where they encounter various metabolic factors, such as the ratio of primary to secondary bile acids and short-chain fatty acids. These factors induce the spores to transition into vegetative cells. The vegetative cells then journey through the gastrointestinal tract, reaching the ileum and cecum, settling in the vacated anaerobic folds. Some of these cells proceed to the colon. In the colon, nutrient deprivation triggers the release of a battery of toxins (tcdA, tcdB, and cdt (CDT)), potentially causing prolonged inflammation (indicated by the red color) of the epithelial layer. Alternatively, the vegetative cells may enter sporulation and are excreted with feces. Without proper sanitation, these spores can adhere to various inanimate and animal surfaces, including clothing, metals, and dogs, thereby perpetuating the infection cycle.

**Figure 2 ijms-25-08672-f002:**
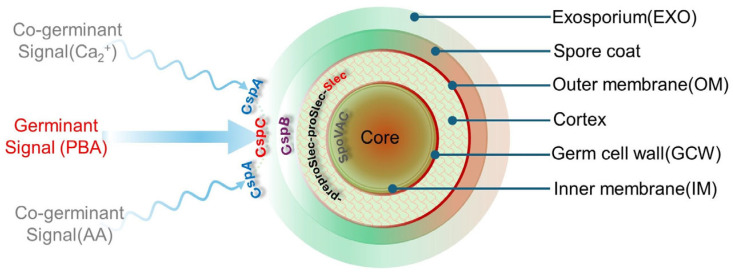
A schematic representation of a *C. difficile* spore structure and its interactions with various germinants (primary bile acids, PBAs) and co-germinates (amino acids, AAs) via key Csp subtilisin-like serine pseudoproteases (CspA and CspC). At the center of the spore, a dehydrated core houses essential germination components (DNA–small acid proteins, Ca^2+^–dipicolinic acid, ribosomes, tDNA), protecting them from UV, chemical, desiccation, and heat damage. An inner membrane and a germ cell wall encase the core. The inner membrane contributes to spore resistance, whilst the peptidoglycan cell wall changes to a cell wall upon germination. Moving outwards from the core, the cortex is composed of cross-linked peptidoglycan, which is subject to degradation by the lytic enzyme SleC following the cleavage of the N-terminal of proSlec by CspB. Although not shown, lytic activity is also regulated by YabC, which converts preproSlec to proSleC. The cortex is surrounded by an outer membrane and a spore coat whose role is to guard against enzyme, biocidal, and chemical attacks. Finally, the exosporium provides another layer of resistance to sterilizing treatments (heat and ethanol).

**Figure 3 ijms-25-08672-f003:**
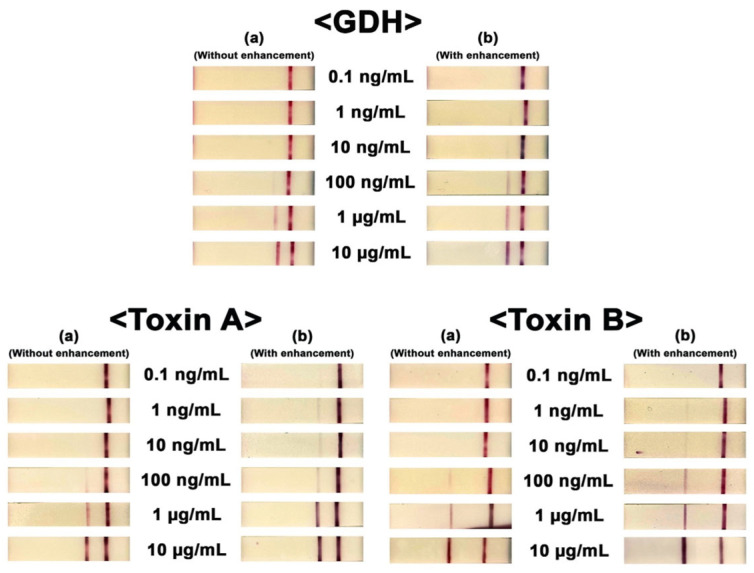
Visual confirmation of the three targets (GDH, toxin A, and toxin B). Concentrations ranged from 0.1 ng/mL to 10 μg/mL. Photo columns (a) and (b) show the results obtained for each test without and with signal enhancement. The figure was modified with permission [90].

**Figure 4 ijms-25-08672-f004:**
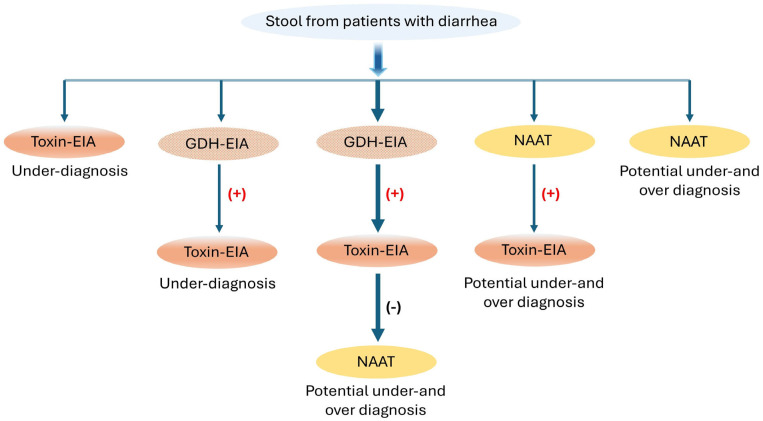
Recommended test algorithms for the diagnosis of CDI incorporating EIA, GDH, and NAAT tests; (+) positive, (−) negative. Modified with permission [36].

**Table 1 ijms-25-08672-t001:** Selection of traditional and emerging diagnostic methods for *C. difficile* detection.

Test Method	Type	Sensitivity	Specificity	Time to Result	Cost	Comments
PCR	Molecular	80–100%	87–99%	Hours	High	Detects toxin genes, may over diagnose due to asymptomatic carriage.
EIA	Immunological	52–75%	96–98%	Hours	Low	Detects toxins A and/or B, low sensitivity.
GDH-EIA	Enzymatic	50–99%	70–100%	Hours	Low	Cannot differentiate between toxigenic and non-toxigenic strains.
TC	Microbiological	90–95%	90–95%	3–5 Days	High	Gold standard, time-consuming, subject to user expertise.
CCNA	Cell-based	90–95%	95–98%	24–48 h	High	Gold standard for toxin detection, time-consuming.
MALDI-TOF MS	Mass Spectrometry	90–100%	90–100%	Minutes	Medium	Rapid identification of strains, effective for epidemiological studies.
LAMP	Molecular	90–100%	94.5–99.1%	Minutes to Hours	Medium	High sensitivity, simpler than PCR, suitable for point-of-care testing.
SIMOA	Molecular /Immunoassay	97.7% (tcdA) 100% (tcdB	100%	Minutes to Hours	High	Automated, detects individual molecules, high performance in identifying true CDI.
Singulex Clarity	Molecular/ Immunoassay	97.7% (tcdA) 100% (tcdB)	100%	<40 min	High	Automated, high specificity and sensitivity, suitable for standalone CDI diagnostics.

## Data Availability

No new data were created or analyzed in this study.

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
