# Peer review of "Emerging Diagnostics in Clostridioides difficile Infection"

_ijms, 2024, doi:10.3390/ijms25168672_

Round 1

Reviewer 1 Report

Comments and Suggestions for Authors

See attached annotated manuscript.

Comments on the Quality of English Language

Author Response

Comment 1. The infinite number of typos and misreferences have been corrected.

Thank you.

Comment 2. The introduction has been reduced. The main text has been rewritten.

Thank you.

A new version submitted.

Thank you.

Reviewer 2 Report

Comments and Suggestions for Authors

The manuscript is relatively well structured but needs improvement in terms of discussion.

The article addresses an important and relevant topic in the medical field, highlighting the established, and emerging technologies used to detect community-acquired Clostridium difficile.

Ultrasensitive tests are difficult to put into practice, so this article has little clinical relevance.

Recommendations for improvement: propose how these tests could be used in clinical practice

Author Response

Comments:

Ultrasensitive tests are difficult to put into practice, so this article has little clinical relevance.

Recommendations for improvement: propose how these tests could be used in clinical practice.

Answers:

The high costs associated with automated RTCA and ultra-sensitive immunoassays(USI) may limit their broader usage in point-of-care settings, particularly when good-quality fecal samples and storage are available[ 113-115].

 Sample quality assured, automated USI may still provide a distinct advantage in detecting early infections or mild cases where the pathogen load is below the detection threshold of standard tests[116]. In addition, USI could play a role in the health of immunocompromised and immunosenescent individuals where low levels of toxin can be clinically significant, permitting timely changes to treatment and vaccination strategies[117-119]. Moreover, regular screening of elderly cohorts in care settings via automated USI could also enhance patient monitoring by accounting for atypical symptoms and comorbidities [120], thereby further reducing the risk of CDI transmission and the importation of new cases.

Reviewer 3 Report

Comments and Suggestions for Authors

Overall, this review is comprehensive. Thus, I just have several minor suggestions.

1. The introduction is too long. Please shorten it to focus on the diagnostics.

2. May add one more figure to briefly summary the diagnostic algorithm of CD diseases based on this review.

3. Please explain the abbreviation in the table 1 in the Note.

Author Response

Comment 1; The introduction is too long. Please shorten it to focus on the diagnostics.

The introduction has been reduced. 

Comment 2; Thank you 

Comment 3; Abbreviations can be found in the updated text